# Replication-dependent size reduction precedes differentiation in *Chlamydia trachomatis*

Jennifer K. Lee [1], Germán A. Enciso[2], Daniela Boassa [3], Christopher N. Chander[1], Tracy H. Lou[1], Sean S. Pairawan[1], Melody C. Guo[1], Frederic Y.M. Wan[2], Mark H. Ellisman[3,4,5], Christine Sütterlin [1] & Ming Tan[6]

*Chlamydia trachomatis* is the most common cause of bacterial sexually transmitted infection. It produces an unusual intracellular infection in which a vegetative form, called the reticulate body (RB), replicates and then converts into an elementary body (EB), which is the infectious form. Here we use quantitative three-dimensional electron microscopy (3D EM) to show that *C. trachomatis* RBs divide by binary fission and undergo a sixfold reduction in size as the population expands. Conversion only occurs after at least six rounds of replication, and correlates with smaller RB size. These results suggest that RBs only convert into EBs below a size threshold, reached by repeatedly dividing before doubling in size. A stochastic mathematical model shows how replication-dependent RB size reduction produces delayed and asynchronous conversion, which are hallmarks of the *Chlamydia* developmental cycle. Our findings support a model in which RB size controls the timing of RB-to-EB conversion without the need for an external signal.

[1] Department of Developmental and Cell Biology, University of California, Irvine, CA 92697-2300, USA. [2] Department of Mathematics, University of California, Irvine, CA 92697-3875, USA. [3] National Center for Microscopy and Imaging Research and Center for Research on Biological Systems, University of California, San Diego, La Jolla, CA 92093, USA. [4] Department of Neurosciences, University of California, San Diego, La Jolla, CA 92093, USA. [5] Salk Institute for Biological Studies, San Diego, CA 92037, USA. [6] Departments of Microbiology and Molecular Genetics, and Medicine, University of California, Irvine, CA 92697-4025, USA. Jennifer K. Lee, Germán A. Enciso and Daniela Boassa contributed equally to this work. Christine Sütterlin and Ming Tan jointly supervised this work. Correspondence and requests for materials should be addressed to C.Süt. (email: suetterc@uci.edu)

The intracellular bacterium *Chlamydia* is responsible for a wide range of highly prevalent infections. *Chlamydia trachomatis* is the most common cause of bacterial sexually transmitted infection and accounts for 60% of all infectious disease cases reported to the CDC[1]. Each year, there are an estimated 131 million new cases of *C. trachomatis* genital infections in the world[2, 3]. *C. trachomatis* is also the etiologic agent of trachoma, a communicable but preventable blindness that affects 40 million people in underdeveloped countries[4]. A related species, *Chlamydia pneumoniae*, is a common cause of community-acquired pneumonia[5].

All members of the *Chlamydia* genus cause an intracellular infection in which there is conversion between two morphologic forms of the bacterium[6, 7]. The elementary body (EB) is the infectious, environmentally stable form that binds and enters an epithelial cell. Within a membrane-bound compartment called the chlamydial inclusion, the EB converts into the larger, metabolically active reticulate body (RB). Beginning at 9–12 h post infection (h.p.i.), RBs divide repeatedly to produce several hundred to a thousand progeny. However, an RB is not infectious and must differentiate into an EB for transmission of the infection to a new host cell. RB-to-EB conversion is first detected at about 24 h.p.i. and occurs asynchronously. This unusual developmental cycle ends at 40–48 h.p.i. with release of EBs to infect new host cells.

The serial conversion between two specialized chlamydial forms raises questions about how these developmental events are regulated. The RB can either be divided into two daughter RBs or convert into an EB, making it the stem cell for RB production and, at the same time, the progenitor of the infectious EB. The signal and control mechanism for this cell fate decision are not known. The EB is functionally analogous to a bacterial spore, or endospore, which is the environmentally stable form of bacteria, such as *Bacillus*. However, an RB is produced by cellular differentiation, unlike an endospore, which is generated by asymmetric cell division of a vegetative cell[8].

A quantitative analysis of the dynamics of RB-to-EB conversion has not been performed. The major limitation has been the size of the chlamydial inclusion, which eventually occupies most of the cytoplasm. Conventional two-dimensional (2D) electron micrographs have provided a sampling of the inclusion and have shown that RB-to-EB conversion occurs in a delayed and asynchronous manner[9, 10]. However, the numbers and relative proportions of RBs and EBs have not been determined over the course of the developmental cycle. Upon re-examination, we have noted that these electron micrographs reveal that the first few RBs in an inclusion are generally larger than the RB population at later times, although this difference has not been remarked upon.

Serial block-face scanning EM (SBEM) is a volume EM technique[11, 12] utilized to study the structure and organization of biological objects. It has been commonly applied in neuroscience for 3D visualization of the nervous system ultrastructure[13, 14] and for circuit reconstructions[11, 12, 15, 16]. More recently it has been used with cell culture models to investigate the 3D architecture of organelles, subcellular structures, DNA, and viral proteins[17–19]. However, this powerful method has not been used to investigate an infection with a pathogenic bacterium.

In this study, we use SBEM to provide a comprehensive quantitative analysis of the intracellular chlamydial infection over time. We show that *C. trachomatis* RBs divide by binary fission and that the resulting RBs are smaller and heterogeneous in size. We incorporate this experimental data into a size control model to account for the delayed and asynchronous nature of RB-to-EB conversion.

## Results

**3D EM analysis of the *Chlamydia* inclusion.** We performed SBEM on monolayers of *C. trachomatis*-infected HeLa cells (Supplementary Fig. 1A), which required up to 277 EM sections to analyze an entire inclusion. In each section, we identified and traced the inclusion membrane and four chlamydial forms (Supplementary Fig. 1B). We readily distinguished RBs from smaller, electron-dense EBs, and separately counted dividing RBs, which are replication intermediates with a constriction that produces a characteristic dumb-bell shape. We also identified a conversion intermediate called the intermediate body (IB), which has a target-like appearance from DNA condensation beginning in its center[7]. Stacks of consecutive 60-nm-thick sections were acquired and subsequently digitally aligned, which allowed individual bacteria to be observed and analyzed in multiple successive sections, increasing the accuracy of identification and physical measurements. On average, an EB was analyzed in 6–7 consecutive EM sections, while an RB was examined in 11–21 EM sections. We then combined all the EM sections computationally into a 3D reconstruction of the inclusion (Fig. 1a, Supplementary Movie 1).

Our analysis provided detailed quantitative information about the *C. trachomatis* inclusion and its developmental forms, but was time and labor intensive. For example, the image acquisition time for each EM section averaged 3 min, but a monolayer of infected cells required up to 500 sections for an average total of 25 h. The outline of each bacterium in every EM section was individually traced in a process called segmentation. Segmentation of EBs was semi-automated because of their uniform size and circular shape. However, RBs required manual segmentation because they varied in size and had an irregular shape. Early inclusions containing few chlamydiae within a small number of EM sections were easily analyzed, but each inclusion from late in the developmental cycle required several days to segment because it contained about a thousand RBs, dividing RBs, IBs, and EBs.

Our 3D EM analysis revealed a non-uniform distribution of chlamydial developmental forms in the inclusion. The proportions of the four forms differed between the entire inclusion and single sections (Fig. 1a, Supplementary Fig. 1C). In particular, sections near the pole of the inclusion overrepresented the percentage of dividing RBs (section 154 in Supplementary Fig. 1C). These findings illustrate the sampling bias inherent to conventional 2D EM analysis, which only examines a small part of the inclusion. In contrast, our 3D EM approach allows a comprehensive, quantitative analysis of the large and heterogeneous chlamydial inclusion.

**Temporal analysis of the chlamydial developmental cycle.** To measure dynamic changes in the numbers, proportions, and sizes of chlamydial developmental forms, we analyzed infected cells over the time course of the intracellular infection (Fig. 1b). For validation, our quantifications compared well with conventional methods for measuring average number of chlamydial genomes per cell by qPCR (Supplementary Fig. 2A) and infectious EBs with a progeny assay (Supplementary Fig. 2B). From counting all chlamydiae in the inclusions analyzed, we calculated a chlamydial doubling time of 1.8 h between 12 and 24 h.p.i., which is similar to published studies[9]. We measured an exponential increase in the mean number of chlamydiae per inclusion from 1.3 at 12 h.p.i. to 1163 at 32 h.p.i. (Fig. 1b). Average RB number increased to a peak of 263 at 32 h.p.i. There were equal proportions of RBs and dividing RBs for all time points up to 32 h.p.i., indicating that RB replication was ongoing and that each RB spent about half of its lifespan undergoing cell division. IBs, as a marker of RB-to-EB conversion, were first detected at 24 h.p.i., when there was an

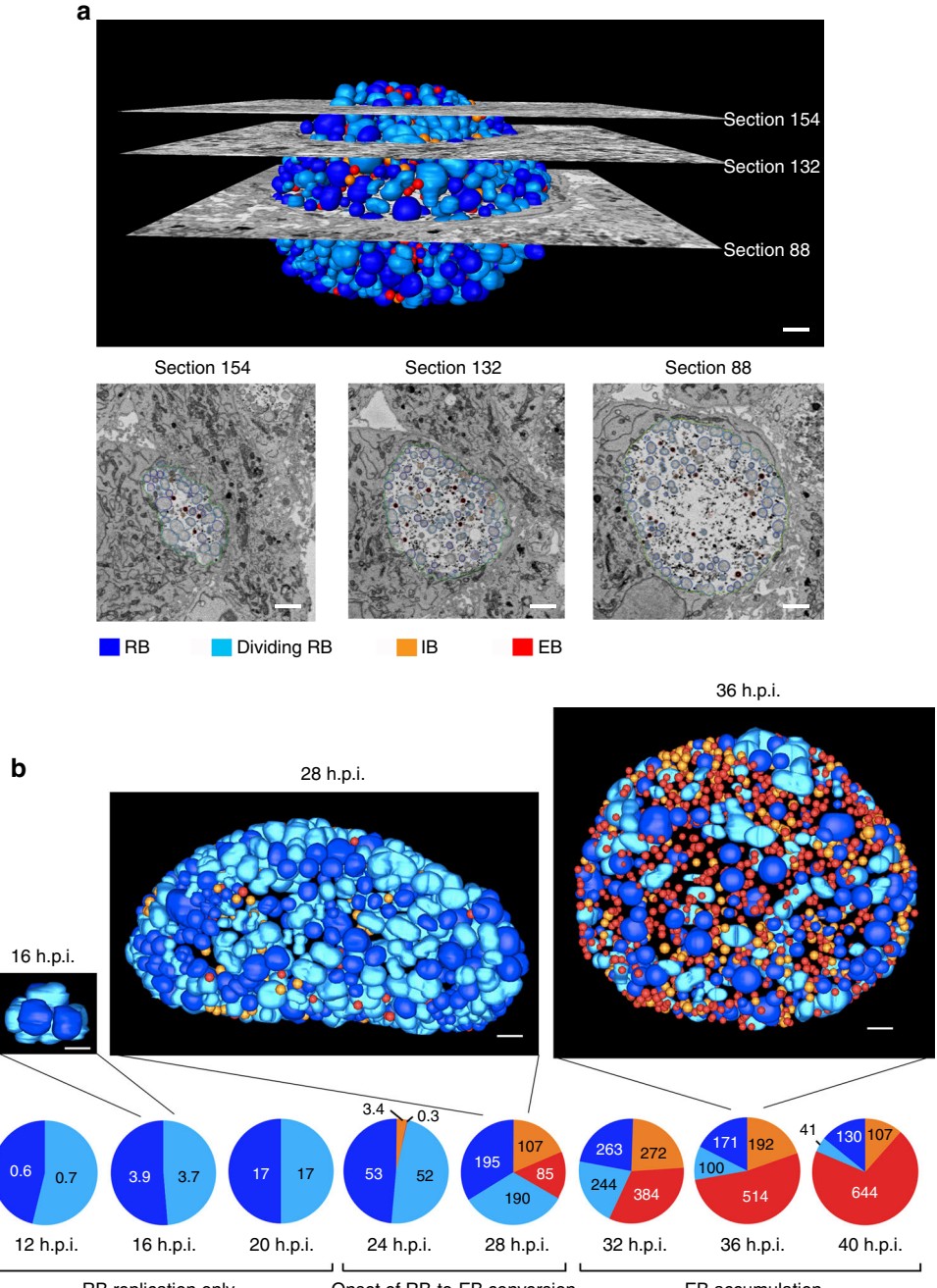

**Fig. 1** Temporal analysis of chlamydial developmental forms using a three-dimensional electron microscopy approach. **a** Serial block-face scanning electron microscopy analysis (SBEM) was used to generate a three-dimensional computational reconstruction of the chlamydial inclusion in a *C. trachomatis*-infected HeLa cell at 28 h post infection (h.p.i.). Micrographs (middle) are shown for sections 154 and 132 (3/4 and halfway up from equator, respectively) and section 88 (equator), with segmentation markings for inclusion membrane (green), RBs (dark blue), dividing RBs (light blue), IBs (orange) and EBs (red). Scale bar: 1000 nm. **b** Entire chlamydial inclusions from representative infected cells at 16, 28, and 36 h.p.i. Scale bar: 1000 nm. Pie charts showing mean numbers of each chlamydial form per inclusion are grouped into three developmental phases: RB replication only (no IBs or EBs), onset of RB-to-EB conversion (IBs + EBs ≤50% of chlamydiae), and EB accumulation (IBs + EBs >50% of chlamydiae). All four chlamydial forms inside each inclusion were identified and counted: 12 h.p.i. (*n* = 50 inclusions), 16 h.p.i. (*n* = 31), 20 h.p.i. (*n* = 22), 24 h.p.i. (*n* = 10), 28 h.p.i. (*n* = 13), 32 h.p.i. (*n* = 10), 36 h.p.i. (*n* = 9), 40 h.p.i. (*n* = 10)

average of 105 RBs, about half of which were visibly dividing. This temporal pattern indicates that at least six rounds of replication had occurred prior to the onset of conversion. At most 23% of chlamydiae were IBs at any time point, emphasizing the asynchronous nature of RB-to-EB conversion. Subsequently there was progressive accumulation of EBs to a maximum of 70% of all

chlamydiae in the inclusion at 40 h.p.i. At late time points, after 32 h.p.i., there were proportionally fewer dividing RBs and IBs, indicating reductions in both RB replication and RB-to-EB conversion.

To investigate how conversion is controlled, we examined if parameters of the inclusion or its bacterial population correlated

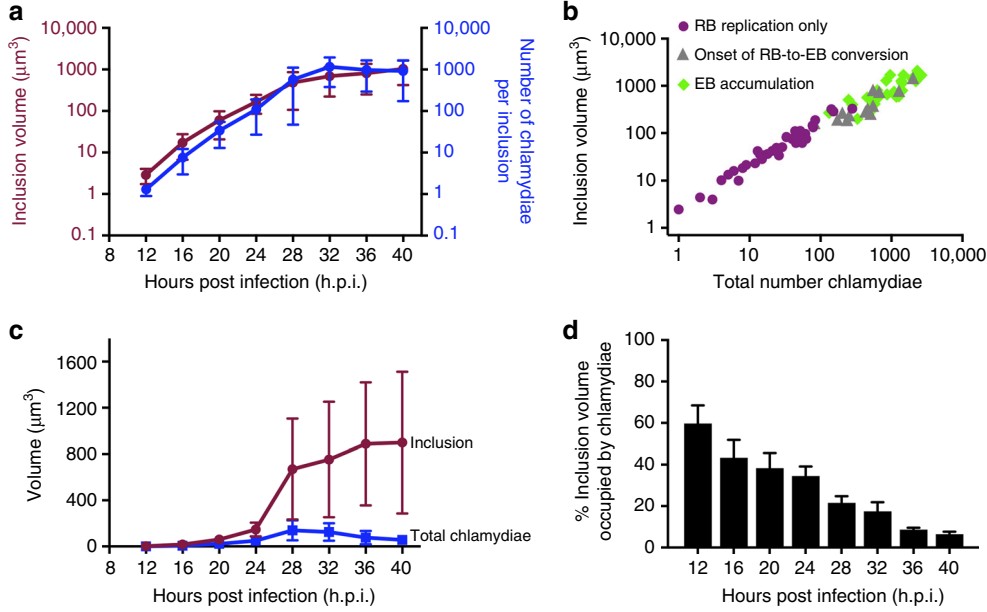

**Fig. 2** Volume analysis of the chlamydial inclusion and chlamydial forms during the developmental cycle. The data presented in this figure is compiled from a total of 155 inclusions: 12 h.p.i. ($n = 50$), 16 h.p.i. ($n = 31$), 20 h.p.i. ($n = 22$), 24 h.p.i. ($n = 10$), 28 h.p.i. ($n = 13$), 32 h.p.i. ($n = 10$), 36 h.p.i. ($n = 9$), 40 h.p.i. ($n = 10$). Error bars represent standard deviation. **a** Temporal change in inclusion volume and number of chlamydiae/inclusion. The data is presented in log scale. **b** Linear relationship between inclusion volume and total number of chlamydiae within that inclusion. Each dot represents a single inclusion, color-coded by its developmental phase, as described in Fig. 1b. **c** Temporal change in inclusion volume and total volume of chlamydiae within the inclusion. **d** % inclusion volume occupied by chlamydiae was calculated for each inclusion as total chlamydial volume divided by inclusion volume

with conversion onset at 24 h.p.i. Mean inclusion volume increased by 300-fold from $2.9\,\mu m^3$ at 12 h.p.i. to $900\,\mu m^3$ at 40 h.p.i., with inclusion growth primarily occurring before 28 h.p.i. (Fig. 2a). However, the density of the chlamydial population remained remarkably constant because the number of chlamydiae in the inclusion increased proportionally with inclusion volume at all times (Fig. 2a, b). In contrast, the total volume of chlamydiae within an inclusion did not keep pace with inclusion growth (Fig. 2c), causing a progressive decrease in the volume fraction of the inclusion taken up by chlamydiae (Fig. 2d). Thus RB-to-EB conversion does not appear to correlate with overall physical crowding in the inclusion, although local crowding effects cannot be excluded.

**Progressive reduction and heterogeneity of RB size.** Unexpectedly, our 3D EM analysis revealed that RBs progressively decrease in size before differentiating into EBs. Mean RB volume decreased from $1.01\,\mu m^3$ (equivalent to a $1.25\,\mu m$ diameter sphere) at 12 h.p.i., to $0.27\,\mu m^3$ at 28 h.p.i. and $0.16\,\mu m^3$ ($0.67\,\mu m$ diameter) at 32 h.p.i., when conversion was actively underway (Fig. 3a). Dividing RBs showed a similar decrease in average size. The finding of concurrent RB replication and size reduction suggests that RBs divide before they double in size.

There was also heterogeneity in RB size within an inclusion. For example, in a single 24 h.p.i. inclusion, the mean size of 40 RBs was $0.37\,\mu m^3$, but 25% were at or below $0.1\,\mu m^3$ with a coefficient of variation (CV, standard deviation/mean) of 97% (Fig. 3c, left panel). In a single 40 h.p.i. inclusion, the mean size of 240 RBs was only $0.21\,\mu m^3$, with 45% at or below $0.1\,\mu m$ and a CV of 118% (Fig. 3c, right panel). For comparison, a steady state population of *Escherichia coli*, which maintains tight size control, had a size distribution CV of 30–40%[20]. This size heterogeneity within a chlamydial population suggests that RB size at division is not tightly controlled.

The lack of size homeostasis in *C. trachomatis* distinguishes it from other bacteria, such as *E. coli*, which maintain cell size

through an "adder model" by adding a constant volume before dividing[21, 22]. In nutrient-poor conditions, bacterial cells get smaller by up to twofold because of lower growth rate[22–24], but the sixfold reduction in RB size during the intracellular chlamydial infection is unprecedented and occurred during exponential growth of the RB population (Supplementary Fig. 2A). Intriguingly, the size ratio of RBs to dividing RBs remained constant at 1.5 from 16 to 40 h.p.i. (Fig. 3b), even as both forms got smaller (Fig. 3a). This stable dividing ratio is reminiscent of the "timer" size control model in *Schizosaccharomyces pombe*, in which there is growth for a fixed time before cell division[25]. However, a *Chlamydia* timer would have to be set to less than the time required for RBs to double in size.

Ongoing size reduction is generally not sustainable, but we surmise that it may be tolerated by chlamydiae because of their unusual developmental cycle. The dramatic and rapid decrease in RB size occurs during a limited number of replication cycles within an infected host cell. Each RB lineage then ends with RB-to-EB conversion instead of being maintained indefinitely. We hypothesize that there is the opportunity to reset RB size when the EB infects a new cell and converts into the initial RB. Thus, size homeostasis may be less critical in *Chlamydia* because it converts back and forth between two developmental forms.

**RBs replicate by binary fission.** Chlamydiae have long been presumed to divide by binary fission[7], but a recent report described polarized cell division in *C. trachomatis*[26]. In that study, confocal and 2D electron microscopy images showed several examples of a smaller, nascent daughter cell budding off an RB. To examine if RBs replicate by binary fission or budding, we used our 3D EM approach to analyze all 114 dividing RBs in two *C. trachomatis* inclusions at 24 h.p.i. Taking the approach used to study *E. coli* cell division, we identified the plane of constriction in each dividing RB to demarcate its two nascent daughter cells. For each daughter, we then calculated the daughter/parent ratio, which is the ratio of the volumes of the

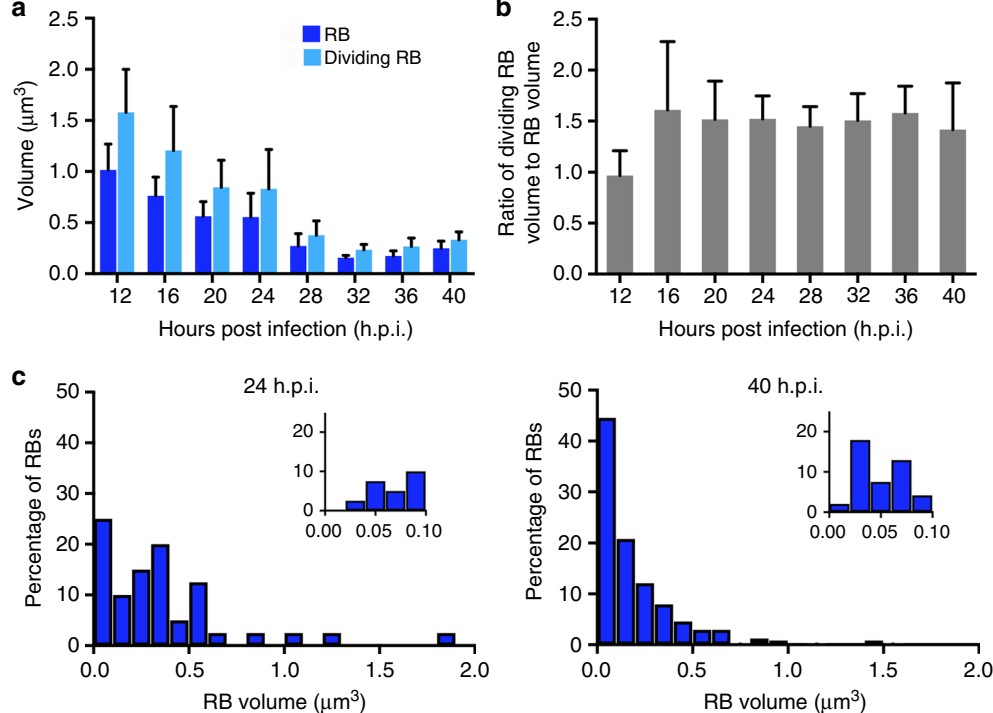

**Fig. 3** RB size decreases and becomes heterogeneous as the developmental cycle progresses. The data for **a** and **b** were compiled from a total of 140 inclusions: 12 h.p.i. ($n = 50$ inclusions), 16 h.p.i. ($n = 31$), 20 h.p.i. ($n = 22$), 24 h.p.i. ($n = 9$), 28 h.p.i. ($n = 7$), 32 h.p.i. ($n = 8$), 36 h.p.i. ($n = 5$), 40 h.p.i. ($n = 8$). Error bars represent standard deviation from the mean. **a** Temporal change in volume of RBs and dividing RBs. Average volume of all RBs or dividing RBs in each inclusion was first determined, and then reported as the mean RB or dividing RB volume for all inclusions at each time point. Mean values are reported and error bars indicate standard deviation. The decrease in RB size was statistically significant between 12 h.p.i. and all later time points (highest p-values were between 12 and 16 h.p.i.: $p = 0.00025$, t-value = 3.9, df = 45, and between 12 and 24 h.p.i.: $p < 0.0001$, t-value = 4.6, df = 28, unpaired t-test). **b** Ratio of dividing RB volume to RB volume during the developmental cycle. For each time point, the ratio was first determined for each inclusion, and then reported as the mean of ratios for all inclusions at that time point. **c** Size histograms for all RBs within a single inclusion at 24 h.p.i. ($n = 40$) and 40 h.p.i. ($n = 240$), distributed into 0.1 μm³ bins. Insets show the smallest bin subdivided into five 0.02 μm³ bins with same y-axis scale

nascent daughter to its dividing RB parent[27]. The CV for the daughter/parent ratio was 11% (Fig. 4), which is only slightly larger than the CV of 4% calculated for *E. coli*[27]. These measurements indicate that RBs divide relatively symmetrically and are consistent with binary fission. We did not detect RBs or dividing RBs with a bud with our 3D imaging method, which should be superior to 2D EM for this task.

We also performed a statistical analysis called the D'Agostino-Pearson test[28] to determine whether polarized cell division is likely in RBs. This approach was used in a classic paper from the 1960s to show that *E. coli* divides by binary fission[27]. If cell division is mediated by binary fission, the distribution of daughter/parent ratios (Fig. 4) approximates a Gaussian curve. Conversely, polarized cell division generates a fraction of daughter cells with very small or very large daughter/parent ratios, which produces a signature graph with a fat tail, and a kurtosis, or "peakness", of the distribution that has a negative value. The distribution of daughter/parent ratios from our analysis of dividing RBs at 24 h.p.i. had a kurtosis of 1.07, which is a positive value that indicates a strong peak and a small tail. This calculation provides quantitative evidence that chlamydiae divide by binary fission rather than by polarized cell division.

**Size control model of RB-to-EB conversion**. What is the significance or function of RB size reduction and size heterogeneity? While decreasing RB size may allow more chlamydiae to fit within the inclusion, there appears to be ample space within late inclusions to accommodate the bacterial population (Fig. 2c, d).

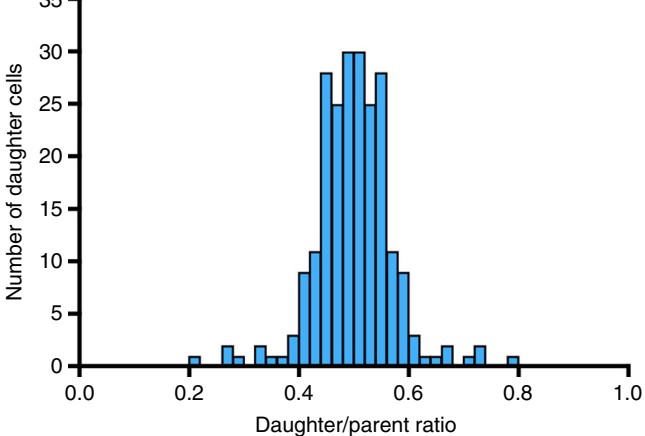

**Fig. 4** RB replication by binary fission. Size histogram of nascent daughter cells at 24 h.p.i. The daughter/parent ratio was calculated as the volume ratio of each daughter cell to its dividing RB parent[27]. Volumes were measured for 228 daughter cells from 114 dividing RBs in two 24 h.p.i. inclusions

Instead, the temporal association between smaller RB size and RB-to-EB conversion suggests that there may be a permissive size for conversion. Based on this observation, we propose a size control mechanism in which RBs decrease in size through replication and can only convert when they reach a minimal size threshold (Fig. 5a). According to this model, the time of

conversion depends on the number of replication cycles required to become small enough to convert. If RB size decreases uniformly in a population, conversion will occur at the same time. However, if there is RB size heterogeneity, as we have detected, a variable number of replication cycles will be required to reach the conversion size, causing conversion to be asynchronous (Fig. 5a).

We designed a stochastic mathematical model to study the ramifications of this size control mechanism (Fig. 5b). We constrained the model with multiple parameters obtained from our studies (Table 1), including direct measurements (initial RB size, time of initial RB replication, variability of daughter cell size in a dividing RB), and calculated values (RB growth rate, mean

transition times from RB-to-dividing RB and from dividing RB-to-2RBs). We also used values that were fit to our data (RB growth rate variability, mean transition times from RB-to-IB and from IB-to-EB). Each RB was allowed to divide at a variable size that was on average less than twice its original size. We estimated an RB threshold size for conversion of 0.06 µm³ for two reasons. This volume is the mean size of an IB when it first appears at 24 h.p.i. (Supplementary Fig. 3A). It is also the size of a very small RB because only 10% of RBs in a 24 h.p.i. inclusion were below this putative size threshold (Fig. 3c).

This stochastic, cell-autonomous model replicates key numerical and temporal features of the chlamydial developmental cycle.

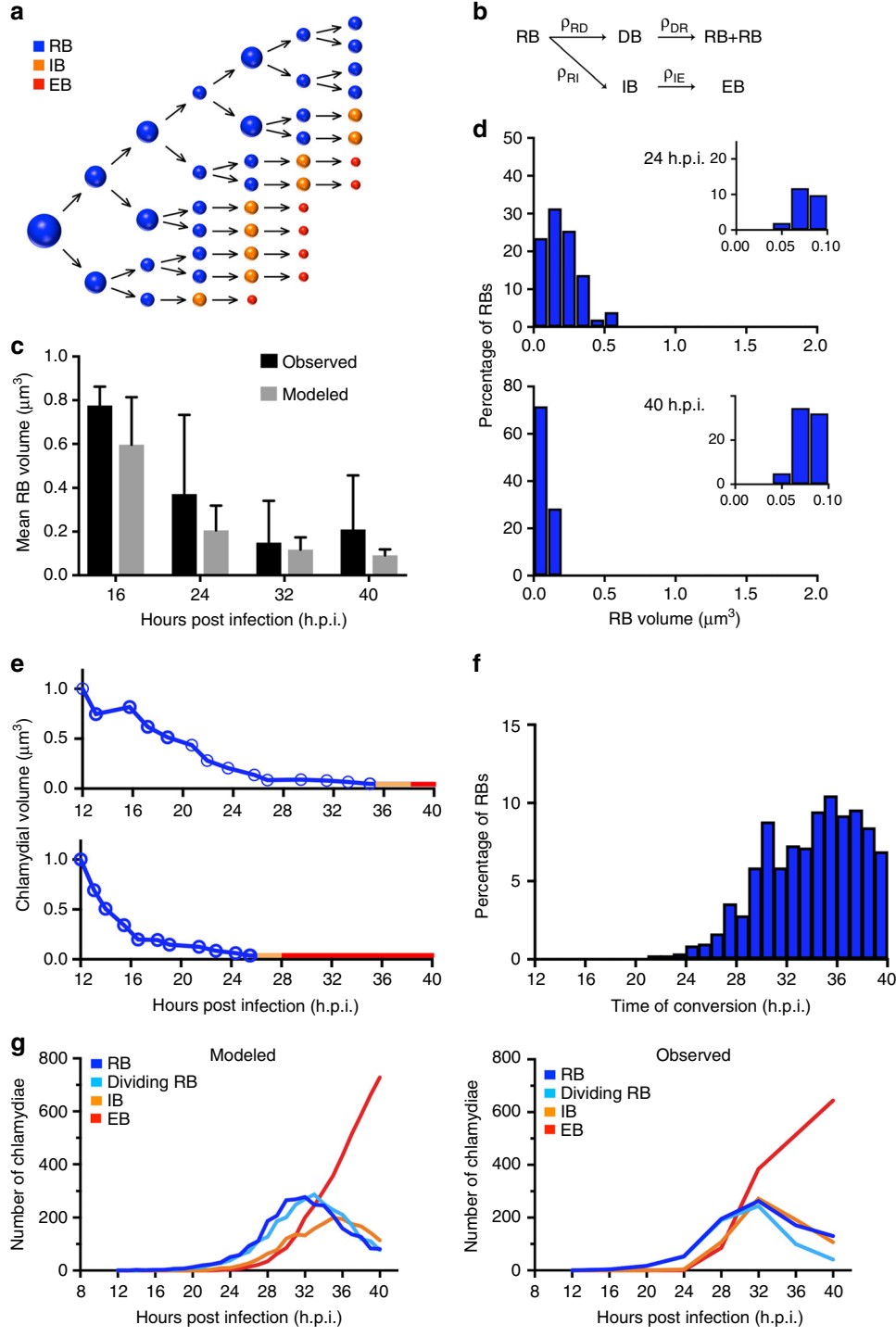

**Table 1 Parameters for the stochastic mathematical model**

| Rate | Symbol | Value | Measurement | Notes |
|---|---|---|---|---|
| Initial RB size | $s_O$ | $1\,\mu m^3$ | Direct | Fig. 3a |
| Initial RB replication time | $t_O$ | 12 h.p.i. | Direct | Fig. 1b |
| DB daughter cell size variability | $\sigma_u$ | 0.05 | Direct | Fig. 3b |
| Mean RB growth rate | $\mu_R$ | $0.25\,h^{-1}$ | Calculated | Figs. 1b, 3b |
| Mean transition time from RB to DB | $\rho_{RD}$ | 0.9 h | Calculated | Fig. 1b |
| Mean transition time from DB to 2RBs | $\rho_{DR}$ | 0.9 h | Calculated | Fig. 1b |
| RB growth rate variability | $\sigma_R$ | $0.04\,h^{-1}$ | Fit to data | Fig. 5 |
| Mean transition time from small RB to IB | $\rho_{RI}$ | 0.1 h | Fit to data | Fig. 5 |
| Mean transition time from IB to EB | $\rho_{IE}$ | 2.5 h | Fit to data | Fig. 5 |
| RB threshold size for conversion | $s_{thr}$ | $0.06\,\mu m^3$ | Inferred | Fig. 3a, Supplementary Fig. 3C |

Parameter values used in the stochastic mathematical model of cell fate regulation through size control. For each parameter we indicate whether its value was directly measured, calculated, or fit so that the model dynamics (Fig. 5) are consistent with experimental data. For instance, the DB daughter cell size variability $\sigma_u$ is measured as the standard deviation of the data in Fig. 3b. The RB threshold size for conversion is inferred from the mean IB size at 24 h.p.i. (Supplementary Fig. 3A), and the size of a small RB (10th percentile) in a 24 h.p.i. inclusion (Fig. 3b)

It produces a progressive decrease in average RB size (Fig. 5c), and size heterogeneity within an inclusion (Fig. 5d). It also reproduces the initial phase of RB replication without EB production, followed by delayed and asynchronous RB-to-EB conversion (Fig. 5e, f). The size control model generated growth curves that most closely resembled those of infected cells (Fig. 5g) when RBs divided at $1.6 \pm 0.3$ times their starting size (Supplementary Note 2). There has been speculation that RB-to-EB conversion is controlled by external signals, such as type III secretion and contact between the RB and the inclusion membrane[29]. However, our stochastic model demonstrates that the observed reduction in RB size could be used as a feedback-independent mechanism to control the dynamics of RB-to-EB conversion. We have not yet been able to test this stochastic model by manipulating the timing of RB replication and RB-to-EB conversion, but automation of the 3D EM segmentation step will help streamline the analysis.

How might RB size control conversion? We speculate that size reduction may facilitate conversion because a decrease in the 25–50-fold size disparity between the initial RB (average volume $1.01\,\mu m^3$, Fig. 3a) and an EB (average volume $0.02$–$0.04\,\mu m^3$, Supplementary Fig. 3B) may make conversion more energetically favorable. The geometry of a smaller RB may also reduce its contact with the inclusion membrane, which has been proposed as a signal that prevents RB-to-EB conversion[29]. Alternatively, a smaller RB could have a higher effective concentration of a conversion-promoting factor.

There is also precedent for an inhibitory DNA-binding factor to be titrated out by the higher DNA-cytoplasmic ratio of a smaller cell[30].

## Discussion

This study provides an unprecedented quantitative analysis of the entire chlamydial inclusion and its changing content of developmental forms over time. Our approach reveals that *C. trachomatis* RBs replicate by binary fission and decrease in size with successive rounds of replication prior to RB-to-EB conversion. This size reduction was detected even though our analysis was limited to discrete time points in the developmental cycle, rather than real-time measurements of bacterial size over multiple RB generations, which is not technically feasible at present for this intracellular infection. We propose that there is a minimal size threshold for RBs to convert. As a consequence, RB size may act as an intrinsic signal to delay conversion until the RB population has expanded. Our stochastic modeling shows that progressive reduction in RB size to a threshold size is sufficient to produce the observed delayed in RB-to-EB conversion without external feedback. Moreover, the asynchronous nature of conversion can be accounted for by variability in both RB size and the number of RB cell divisions to reach the threshold size. This size control model bears resemblance to midblastula transition, in which there are multiple rounds of cell division before a developmental switch during early embryogenesis[30]. Our finding of RB size reduction

**Fig. 5** Analysis of size-dependent control of RB-to-EB conversion using a stochastic mathematical model. **a** Proposed model in which the size of an RB determines whether it can convert or continues to replicate. RBs become progressively smaller because they divide, on average, at less than twice their starting size. They can only convert into an EB below a permissive size. These two elements of RB size control ensure that the RB population expands before conversion occurs. The figure demonstrates how weak control of RB size at replication can produce size heterogeneity and lead to asynchronous conversion by varying the number of replication cycles required to reach the conversion size threshold. **b** Wiring diagram to show the four different variables in the system and the four possible transformations. Details of the mathematical model provided in Supplementary Notes. **c** Mean volume of the RB population within an individual inclusion, measured experimentally (16 h.p.i. $n = 8$ RBs, 24 h.p.i. $n = 40$, 32 h.p.i. $n = 245$, 40 h.p.i. $n = 240$), or produced by the size control model, at selected time points. Error bars indicate standard deviation. **d** Histograms of RB size obtained with the mathematical model for single inclusions at 24 and 40 h.p.i. recapitulate the experimental data in Fig. 3c. **e** Two sample time courses from the model illustrating how different RB lineages culminate in different times of RB-to-EB conversion. Each time course consists of successive rounds of RB replication (blue line) followed by conversion to an IB (orange line) and then EB (red line). Each newly produced RB shown by an open circle. **f** Histogram showing time of RB-to-EB conversion predicted by the mathematical model for all EBs produced in a single inclusion by 40 h.p.i. **g** Growth curves showing the mean number of each chlamydial form/inclusion. The graph on the left was produced by the stochastic size control model, while the graph on the right shows growth curves from the 3D EM analysis of *Chlamydia*-infected cells (12 h.p.i. $n = 50$ inclusions, 16 h.p.i. $n = 31$, 20 h.p.i. $n = 22$, 24 h.p.i. $n = 10$, 28 h.p.i. $n = 13$, 32 h.p.i. $n = 10$, 36 h.p.i. $n = 9$, 40 h.p.i. $n = 10$)

before differentiation suggests that *C. trachomatis* may also use its cell size as a developmental timer.

## Methods

**Cell culture and *Chlamydia* infections.** HeLa cells (ATCC CCL-2) were grown in Advanced DMEM (4.5 g glucose per L) (Invitrogen) supplemented with 2% fetal bovine serum (FBS) (Hyclone/Thermo Fisher) and 2 mM GlutaMAX-I (Invitrogen) in 5% $CO_2$ at 37 °C.

The cell monolayers were infected with *C. trachomatis* serovar L2, strain L2/434/Bu (ATCC VR-902B) at a multiplicity of infection (MOI) of 3 in sucrose-phosphate-glutamic acid (SPG). Uninfected control experiments were performed as mock infections in SPG alone. Infections were carried out by centrifugation at 700×*g* in a Sorvall Legend Mach 1.6 R centrifuge for 1 h at room temperature. After centrifugation, the inoculum was replaced by fresh cell culture medium and monolayers were incubated at 37 °C and 5% $CO_2$. HeLa cells and EBs were verified to be free of *Mycoplasma* contamination by PCR[31].

**Preparation of cells for serial block-face scanning EM.** *Chlamydia*-infected monolayers were fixed in a solution of 2% paraformaldehyde and 2.5% glutaraldehyde in 0.1 M cacodylate buffer, pH 7.4 for 1 h. The cells were stained for SBEM as previously reported[17]. Briefly, cells were washed 5X in cold 0.1 M cacodylate buffer then incubated in solution containing 1.5% potassium ferrocyanide and 2% osmium tetroxide supplemented with 2 mM calcium chloride in 0.1 M cacodylate buffer for 30 min on ice. After 5 × 2-min washes in doubled distilled water, cells were incubated in 1% thiocarbohydrazide for 10 min at room temperature. Following 5 × 2-min washes in double distilled water at room temperature, cells were placed in 2% osmium tetroxide in double distilled water for 10 min at room temperature. The cells were rinsed 5 × 2 min with double distilled water at room temperature and subsequently incubated in 2% uranyl acetate at 4 °C overnight. The next day, cells were washed 5 × 2 min in double distilled water at room temperature and en bloc Walton's lead aspartate staining was performed for 10 min at 60 °C. Following 5 × 2-min washes in double distilled water at room temperature, cells were dehydrated using a series of ice-cold graded ethanol solutions and then embedded in Durcupan ACM resin (Electron Microscopy Sciences). The resin was allowed to polymerize in a vacuum oven at 60 °C for 48 h. SBEM imaging was completed using a Gatan automated 3View system (Gatan Inc.) incorporated into a Zeiss Sigma or Merlin Compact Scanning Electron Microscope (Zeiss), and images were recorded at 60 nm cutting intervals. For details on image sizes of micrographs produced from each *Chlamydia*-infected monolayer, Supplementary Table 1.

**3D EM segmentation and analysis.** Complete three-dimensional reconstructions of *Chlamydia* inclusions were constructed and analyzed using the IMOD image processing software (University of Colorado, Boulder). Inclusion membrane and chlamydial forms were marked on 2D electron micrographs then assembled together to build the 3D models. Numerical and volumetric analyses were conducted using plug-ins of the IMOD software (3Dmod).

3D models were reconstructed for the inclusion in each of 155 *Chlamydia*-infected cells. Fifty inclusions at 12 h.p.i. and 31 inclusions at 16 h.p.i. were analyzed. Because of the labor-intensive nature of segmentation, at least 9 inclusions were analyzed at later time points (4-h intervals between 20 and 40 h.p.i.) when inclusions were large and contained many chlamydiae. For each of these later time points, representative inclusions were selected by predetermining the volume of >20 inclusions, sorting them by size into three bins (large, medium, and small), and using a random number generator to select at least 3 inclusions per bin for analysis (Supplementary Table 2).

**Analysis of RB cell division.** A total of 114 dividing RB from two 24 h.p.i. inclusions were analyzed in multiple EM sections of 0.06 μm thickness. For each dividing RB, the plane of constriction[27] was identified so that each of the two nascent daughter cells could be demarcated and segmented. The volumes of the parent dividing RB and each daughter cell were then determined from the 3D reconstruction, and a ratio of each daughter volume/parent volume was calculated.

**Mathematical modeling.** A continuous-time, stochastic model of cell-size dynamics was designed using parameters based on experimental data from this study. The transitions from RB to dividing RB (DB), RB to IB, and IB to EB follow the network described in Fig. 5b for the population model, with gamma distribution for each transition time. In addition, RB-to-IB conversion can only occur when size decreases below a specified threshold. In this model, the size of each chlamydia at any given time is defined. The exponential growth rates of RBs and DBs are chosen independently from a normal distribution after each transition. The size of a daughter RB is determined from the size of its mother DB at the time of division, using a binomial partitioning method that introduces randomness in the division. The size of an IB and its successor EB is determined by the size of its RB progenitor before conversion. EB size remains constant after conversion. All modeling was carried out using Matlab; see Supplementary Note 2 for additional details on this model and Supplementary Note 3 for parameter values. The parameter values of the model are based on experimental data, mostly from direct

measurements or computed from experimental values. See Table 1 for the parameters used, as well as their values and the form of derivation.

**Statistical information.** For Figs 1b, 5g, Supplementary Fig. 2, all of the four chlamydial forms inside 155 inclusions were identified and counted: 12 h.p.i. (*n* = 50 inclusions), 16 h.p.i. (*n* = 31), 20 h.p.i. (*n* = 22), 24 h.p.i. (*n* = 10), 28 h.p.i. (*n* = 13), 32 h.p.i. (*n* = 10), 36 h.p.i. (*n* = 9), 40 h.p.i. (*n* = 10). Figure 2 reports the mean volume of these 155 inclusions by time point. Figures 2c, d, 3a, b and Supplementary Fig. 3 present a more time-intensive measurement of mean volume for each of the four chlamydial forms within 140 inclusions: 12 h.p.i. (*n* = 50 inclusions), 16 h.p.i. (*n* = 31), 20 h.p.i. (*n* = 22), 24 h.p.i. (*n* = 9), 28 h.p.i. (*n* = 7), 32 h.p.i. (*n* = 8), 36 h.p.i. (*n* = 5), 40 h.p.i. (*n* = 8). For Fig. 4, all 114 dividing RBs from two inclusions at 24 h.p.i. were analyzed. Figure 5c is an analysis of mean RB volume for the entire RB population within a single inclusion at 16 h.p.i. (*n* = 8 RBs), 24 h.p.i. (*n* = 40), 32 h.p.i. (*n* = 245), and 40 h.p.i. (*n* = 240). Error bars in all graphs represent standard deviation from the mean.

For Fig. 3a, the progressive decrease in RB size was analyzed with an unpaired *t*-test and found to be statistically significant between 12 h.p.i. and each of the later time points, e.g., 12 h.p.i. and 16 h.p.i.: $p = 0.00025$, *t*-value = 3.9, df = 45; 12 h.p.i. and 20 h.p.i.: $p < 10^{-7}$, *t*-value 7.0, df = 39; 12 h.p.i. and 24 h.p.i.: $p < 0.0001$, *t*-value = 4.6, df = 28.

**Data availability.** All SBEM image data can be accessed by downloading from the Cell Centered Database and Cell Image Library under project ID 20099 (http://www.cellimagelibrary.org/images?k=project_20099&simple_search=Search). Other relevant data supporting the findings of the study are available in this article and its Supplementary Information files, or from the corresponding authors upon request.

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

## Acknowledgements

We thank Arthur Lander, Bert Semler, Eric Stanbridge and Steven Gross (all from the University of California, Irvine) for critical reading of the manuscript and helpful comments and A. Noske for technical help with 3D EM segmentation and analysis. This work was supported by a grant from the NIH (R21 AI117463) (M.T.) and a Research Scholar Grant from the American Cancer Society (C.S.). J.K.L. was supported by a training grant from the National Cancer Institute (T32 CA0090054). This work was also supported by a NIH grant to M.H.E. (P41 GM103412) for support of the National Center for Microscopy and Imaging Research.

## Author contributions

J.K.L. conducted the experiments and analyzed 3D EM data; G.A.E. and F.W. designed, performed, and analyzed mathematical modeling simulations and wrote the mathematical analysis; D.B. prepared 3D EM samples and collected and processed micrograph data; M.H.E. was involved in study design and data analysis for the EM work; C.N.C., T. H.L., S.S.P., and M.C.G. segmented 3D EM data; C.S., M.T., and G.A.E. performed the study design and analyses; M.T. and C.S. wrote the manuscript with assistance from J.K. L., G.A.E., and D.B.

## Additional information

**Competing interests:** The authors declare no competing financial interests.

