## [Peer Review File · Nature Communications]

Reviewers' comments:

Reviewer #1 (Remarks to the Author):

This manuscript presents a quantitative structural analysis of the entire chlamydial inclusion as a function of time when human-derived HeLa cells are infected with these important pathogenic bacteria. To obtain these results, the authors use the technique of serial block-face electron microscopy (SBEM), which is unique in being able to image large fields of view (>20 μm) while maintaining a spatial resolution (or pixel size) of a few nanometers. The authors demonstrate that a complete structural analysis of the chlamydial inclusions can only be obtained by analyzing entire 3D volumes, which provides information about the size and shape of the inclusions at different times post-infection, and also allows a comprehensive determination of the shapes and sizes of all the bacteria in the inclusions (which can be in excess of 1,000). The bacteria are found to undergo large changes in size as they make a transition from reticular body to elemental body, which is the infectious particle.

In the opinion of this reviewer, the authors succeed in their goal of tracking a medically important infection process through 3D space and time by using this novel structural approach, i.e., SBEM. The authors have performed an extensive study and have included a thorough statistical analysis.

Minor comments:

1. To make the manuscript more accessible to the general reader, who might not be an expert in microbiology, it might be useful to include a few more sentences of introduction to mention that the EB is somewhat equivalent to a spore.
2. Please could the authors indicate the pixel size (x and y), and the total acquisition time needed to collect data for each inclusion. Also, it would be highly beneficial for the reader to know how much time was taken to segment each of the larger inclusions, as well as the time required to build and analyze the models. Such information will inform the reader about the scale of the work.

Reviewer #2 (Remarks to the Author):

Review: Replication dependent size reduction precedes differentiation in Chlamydia

This manuscript by Lee and colleagues uses serial block face scanning EM to perform a quantitative analysis of *Chlamydia trachomatis* inclusion contents throughout its developmental cycle. Through these analyses and observations, they propose a novel mechanism which suggests a size dependent conversion event that *Chlamydia* may employ to transition from its replicative (RB) form to its infectious (EB) form within an inclusion.

This is a potentially paradigm establishing mechanism/model. The relative abundance and general morphology of the inclusion contents (RB/EB) throughout the developmental cycle are poorly understood as are the mechanisms that *Chlamydia* may utilize to control this differentiation. As such, these findings have high significance and impact for the *Chlamydia* field.

The overall observations are compelling. It appears evident that the size of RBs – replicating or individual – reduced with the continuation of the developmental cycle (12- 32 hpi). Infectious EBs were morphologically detected after 24 hpi (Fig1B, S2B, and S3) and correlated with the presence/amount of infectious, inclusion forming, EBs (Fig 2B). The measurement of the reduction indicates a threshold that is reached before conversion and that the diversity in RB size and

division provide the asynchrony associated with Chlamydia conversions.

One of the major concerns with the manuscript is the disjointed nature of data presentation and overall explanations. The major observations and key points can be 'dug out' from the various sections (sup/fig legends/text) but it is not at all a fluid description. Moreover, a couple of very key points need to be at the forefront of explanation – such as the threshold (0.06 cub mic) that is apparently reached for conversion and that the asynchronous conversion is provided by the variability in RB sizes AND division reductions. This information is in the manuscript, but buried in various sections and requires that a reader pull together these from various paragraphs.

This (size reduction for differentiation) is an intriguing hypothesis based upon these observations; however, the observational analysis was performed on only one strain of *C. trachomatis* (LGV). There are growth and phenotypic differences between closely related species (e.g. *C. muridarum* and *C. trachomatis* serovar A or D). Due to the potentially paradigm establishing observation, analysis on other strain and species with different growth rates is of importance. This becomes more relevant as the title of the manuscript states "Chlamydia". Experiments performed on other species and strains would solidify this claim. Alternatively, a fairly strong 'disclaimer' could be included in this manuscript that emphasizes the singular strain/species and limitations.

There are no experimental studies performed (no variables changed). The analysis is purely observational and used to support a hypothesis which is not really tested in this manuscript. While this is certainly acceptable, this 'limitation' may be considered for inclusion in the manuscript.

Recent publication has reported the possibility that Chlamydia divide via polar division (in contrast to binary/middle plane septum formation). The authors should, at minimum, discuss the presence, or absence, of these polar division events. This high-resolution analysis would certainly be expected to observe these forms. Moreover, it may be very relevant to a smaller daughter cell and the general reduction of bacterial cells.

The authors make a statement that "Thus RB-to-EB conversion does not appear to correlate with physical crowding in the inclusion." It would be best (for full clarity) to also state at this point in the manuscript that growth limitations could be linked with organisms associated at the inclusion membrane. The discussion enables this consideration but it seems best to also enable consideration at this point in the manuscript.

The authors state that "This size heterogeneity indicates that RB size at division is not tightly controlled." This is based upon measurements of RBs at different times and the variation that is present (CV of 97% and 118%). It would be most useful to compare this division variation to other bacteria to support this claim. *E. coli* and *Bacillus* are obvious comparisons, but others with unique growth phenotypes (e.g. *Corynebacterium*) may be useful as well.

While the model proposing a size dependent conversion event is an intriguing concept, there may be an additional and intuitive reason for the step-wise reduction in RB size after division, serving two main purposes: 1) to increase the overall number of RBs while maintaining a manageable (within host cell) inclusion size, and 2) decrease the size of RBs in order to make the transition into the smaller EB form more seamless and not requiring catabolism or blebbing of excess membrane.

Reviewer #3 (Remarks to the Author):

The paper studies dynamics of the *Chlamydia trachomatis* which is an important problem since

Chlamydia trachomatis accounts for 60% of infectious disease cases reported in the U.S. This type of infection converts reticulate body (RB) into the infectious elementary body (EB) inside a eukaryotic host cell for transmission to a new host cell. The authors indicate that signal or control mechanism for RB-to-EB conversion is not known. Also, the numbers and relative proportions of RBs and EBs in an inclusion have not been determined during the developmental cycle. This is why they used a three-dimensional electron microscopy (3D EM) method, known as serial block-face scanning EM (SBEM), to visualize the entire chlamydial inclusion. They were able to demonstrate, based on analysis of the obtained imaging data, that at least six rounds of the RB replication occurred prior to the onset of conversion which is an interesting and potentially important biologically relevant observation. They also observed that RBs progressively decreased in size after each replication before differentiating into EBs. Based on this observation, the authors hypothesize a size control mechanism in which RBs decrease in size through replication and only convert when they reach a minimal size threshold. The authors used a macro-scale population level stochastic mathematical model as well as classical size-structured model to test this hypothesis.

Unfortunately, the model uses many simplifying assumptions including symmetry assumption, which limit its predictive ability at micro-scale. There is no biological justification provided for using gamma distribution and values of many parameters in the model are not linked to biological observables at micro-scale sub RB and EB level. For example, the authors just state that "Rather than an exponential distribution as is common for chemical reactions, we use a gamma distribution, which represents the time before a given number of events in a Poisson processes, and which better describes the timing between cell transitions" without providing any justification. Also, it is not clear how parameters can be determined using experimental observations for the normal distribution used for determining the growth rate in the context of the cell size dynamics.

It is not clear how the model can be used to test suggestion by the authors that "conversion could be actively promoted in a smaller RB by the higher effective concentration of a positive regulator" since it does not describe molecular mechanisms of conversion. Also, authors suggest that "contact between the RB and the inclusion membrane, which has been proposed to prevent RB-to-EB conversion, could be reduced by the geometry of a smaller RB". At the same time, their model does not describe in detail shapes of RBs and membranes as well as interactions of RBs with membranes.

To summarize, the paper describes novel potentially important biologically relevant observations. At the same time, proposed oversimplified models do not provide clear explanation of how exactly RBs and host cells could implement feedback-independent mechanism based on the RB size to control the dynamics of RB-to-EB conversion.

We would like to thank the reviewers for their insightful comments on our manuscript. In response have addressed all of the reviewers' points of criticism and concerns as outlined below. Changes to our manuscript are described below and highlighted in the text in yellow.

Reviewer #1 (Remarks to the Author):

This manuscript presents a quantitative structural analysis of the entire chlamydial inclusion as a function of time when human-derived HeLa cells are infected with these important pathogenic bacteria. To obtain these results, the authors use the technique of serial block-face electron microscopy (SBEM), which is unique in being able to image large fields of view (>20 μm) while maintaining a spatial resolution (or pixel size) of a few nanometers. The authors demonstrate that a complete structural analysis of the chlamydial inclusions can only be obtained by analyzing entire 3D volumes, which provides information about the size and shape of the inclusions at different times post-infection, and also allows a comprehensive determination of the shapes and sizes of all the bacteria in the inclusions (which can be in excess of 1,000). The bacteria are found to undergo large changes in size as they make a transition from reticular body to elemental body, which is the infectious particle.

In the opinion of this reviewer, the authors succeed in their goal of tracking a medically important infection process through 3D space and time by using this novel structural approach, i.e., SBEM. The authors have performed an extensive study and have included a thorough statistical analysis.

Minor comments:

1. To make the manuscript more accessible to the general reader, who might not be an expert in microbiology, it might be useful to include a few more sentences of introduction to mention that the EB is somewhat equivalent to a spore.

We have added a description of the EB in the introduction, including a comparison of the similarities to and differences from a spore, as suggested by the reviewer (p3, lines 11-13).

2. Please could the authors indicate the pixel size (x and y), and the total acquisition time needed to collect data for each inclusion. Also, it would be highly beneficial for the reader to know how much time was taken to segment each of the larger inclusions, as well as the time required to build and analyze the models. Such information will inform the reader about the scale of the work.

We thank the reviewer for this good suggestion to make clear the amount of manual and computational time that the analysis required. Details have been added to the text (p4-5). We have also included an additional table that describes all the data collection parameters (Table 1). Pixel size information has been added to Supplementary Table 1.

Reviewer #2 (Remarks to the Author):

This manuscript by Lee and colleagues uses serial block face scanning EM to perform a quantitative analysis of Chlamydia trachomatis inclusion contents throughout its

developmental cycle. Through these analyses and observations, they propose a novel mechanism which suggests a size dependent conversion event that Chlamydia may employ to transition from its replicative (RB) form to its infectious (EB) form within an inclusion.

This is a potentially paradigm establishing mechanism/model. The relative abundance and general morphology of the inclusion contents (RB/EB) throughout the developmental cycle are poorly understood as are the mechanisms that Chlamydia may utilize to control this differentiation. As such, these findings have high significance and impact for the Chlamydia field.

The overall observations are compelling. It appears evident that the size of RBs – replicating or individual – reduced with the continuation of the developmental cycle (12-32 hpi). Infectious EBs were morphologically detected after 24 hpi (Fig1B, S2B, and S3) and correlated with the presence/amount of infectious, inclusion forming, EBs (Fig 2B). The measurement of the reduction indicates a threshold that is reached before conversion and that the diversity in RB size and division provide the asynchrony associated with Chlamydia conversions.

One of the major concerns with the manuscript is the disjointed nature of data presentation and overall explanations. The major observations and key points can be 'dug out' from the various sections (sup/fig legends/text) but it is not at all a fluid description. Moreover, a couple of very key points need to be at the forefront of explanation – such as the threshold (0.06 cub mic) that is apparently reached for conversion and that the asynchronous conversion is provided by the variability in RB sizes AND division reductions. This information is in the manuscript, but buried in various sections and requires that a reader pull together these from various paragraphs.

We thank the reviewer for this point. For easy reference we have added a paragraph in the introduction that summarizes the qualitative information that we obtained from our studies. A new table was also added that lists all the parameters used in our stochastic mathematical model and the figures in which the values are reported (Table 1). The major observations and key points are also summarized in the last paragraph of the text (p11-12).

This (size reduction for differentiation) is an intriguing hypothesis based upon these observations; however, the observational analysis was performed on only one strain of *C. trachomatis* (LGV). There are growth and phenotypic differences between closely related species (e.g. *C. muridarum* and *C. trachomatis* serovar A or D). Due to the potentially paradigm establishing observation, analysis on other strain and species with different growth rates is of importance. This becomes more relevant as the title of the manuscript states "Chlamydia". Experiments performed on other species and strains would solidify this claim. Alternatively, a fairly strong 'disclaimer' could be included in this manuscript that emphasizes the singular strain/species and limitations.

At the reviewer's suggestion, we have changed the title of the manuscript to make clear that the analysis is limited to *C. trachomatis*. So far, the analysis has not been performed on other *Chlamydia* species because of the time and labor intensive nature of this innovative 3D EM analysis.

There are no experimental studies performed (no variables changed). The analysis is

purely observational and used to support a hypothesis which is not really tested in this manuscript. While this is certainly acceptable, this 'limitation' may be considered for inclusion in the manuscript.

The reviewer is correct that we did not test our size control model with experimental studies, something we are interested in pursuing in future studies. These studies may be challenging because we will have to find ways to change fundamental properties of the chlamydial developmental cycle. However, as a powerful alternative approach, we developed and applied a stochastic mathematical model to test the implications of our size control model and used it to show that RB cell size alone can reproduce the delayed timing and asynchronous onset of RB-to-EB conversion without the need for any external signals. This feedback-independent model is conceptually new for the field and helps to show the potential significance of the RB size reduction that we have discovered. However, testing this feedback-independent size control model experimentally would clearly go beyond the scope of this current study.

Recent publication has reported the possibility that Chlamydia divide via polar division (in contrast to binary/middle plane septum formation). The authors should, at minimum, discuss the presence, or absence, of these polar division events. This high-resolution analysis would certainly be expected to observe these forms. Moreover, it may be very relevant to a smaller daughter cell and the general reduction of bacterial cells.

In response to the reviewer's excellent suggestion, we have added a new section where we have performed an analysis of dividing RBs to determine whether they divide centrally or in a polar manner (p8-9). We have included a new Fig. 4 displaying the experimental data. Moreover, we have analyzed the distribution of daughter volume fractions, following a classic paper that ruled out *E. coli* polar fission in the 1960s. By calculating the kurtosis of the experimental values, we show that the number of very small or very large daughter cells is not significant and that division is therefore likely to be via binary fission rather than polarized cell division.

We also report in this new section that we did not observe the budding phenomenon described by Abdelrahman et al. A significant difference between the two studies is that our 3D EM analysis was based on multiple slices through each individual bacterium which would have allowed the detection of a bud if present. In contrast, the "budding" study was based on 2D EM slices that may not have been through the plane of RB cell division.

The authors make a statement that "Thus RB-to-EB conversion does not appear to correlate with physical crowding in the inclusion." It would be best (for full clarity) to also state at this point in the manuscript that growth limitations could be linked with organisms associated at the inclusion membrane. The discussion enables this consideration but it seems best to also enable consideration at this point in the manuscript.

We have clarified our statement about overall bacterial density in the inclusion to leave open the possibility of local crowding effects (p7, lines 7-8).

The authors state that "This size heterogeneity indicates that RB size at division is not tightly controlled." This is based upon measurements of RBs at different times and the variation that is present (CV of 97% and 118%). It would be most useful to compare this

division variation to other bacteria to support this claim. *E. coli* and *Bacillus* are obvious comparisons, but others with unique growth phenotypes (e.g. *Corynebacterium*) may be useful as well.

We agree with the reviewer about the benefit of comparing the values of other bacteria to make clear the unusual heterogeneity in chlamydial cell size. We have cited an *E. coli* study that reported a much lower cell size CV of 30-40% (p7, lines 19-21).

While the model proposing a size dependent conversion event is an intriguing concept, there may be an additional and intuitive reason for the step-wise reduction in RB size after division, serving two main purposes: 1) to increase the overall number of RBs while maintaining a manageable (within host cell) inclusion size, and 2) decrease the size of RBs in order to make the transition into the smaller EB form more seamless and not requiring catabolism or blebbing of excess membrane.

We have included a paragraph in which we speculate on possible reasons for this size-dependent control mechanism (p11).

Reviewer #3 (Remarks to the Author):

The paper studies dynamics of the *Chlamydia trachomatis* which is an important problem since *Chlamydia trachomatis* accounts for 60% of infectious disease cases reported in the U.S. This type of infection converts reticulate body (RB) into the infectious elementary body (EB) inside a eukaryotic host cell for transmission to a new host cell. The authors indicate that signal or control mechanism for RB-to-EB conversion is not known. Also, the numbers and relative proportions of RBs and EBs in an inclusion have not been determined during the developmental cycle. This is why they used a three-dimensional electron microscopy (3D EM) method, known as serial block-face scanning EM (SBEM), to visualize the entire chlamydial inclusion. They were able to demonstrate, based on analysis of the obtained imaging data, that at least six rounds of the RB replication occurred prior to the onset of conversion which is an interesting and potentially important biologically relevant observation. They also observed that RBs progressively decreased in size after each replication before differentiating into EBs. Based on this observation, the authors hypothesize a size control mechanism in which RBs decrease in size through replication and only convert when they reach a minimal size threshold. The authors used a macro-scale population level stochastic mathematical model as well as classical size-structured model to test this hypothesis.

Unfortunately, the model uses many simplifying assumptions including symmetry assumption, which limit its predictive ability at micro-scale.

We thank the reviewer for this comment – in response to this concern as well as that of another reviewer, we have carried out additional measurements of the existing data to calculate Fig. 4, describing the daughter-parent volume fractions for 114 dividing RBs in two inclusions (p8-9). This information has been used to inform the model on the extent of asymmetric division in the system, which has been incorporated into the current manuscript. We have used a binomial partition mechanism for cell division, which we justify using references from the literature. The standard deviation in this partition has been fit to that of the experimental data, and all calculations are shown in the

Mathematical Supplement.

There is no biological justification provided for using gamma distribution and values of many parameters in the model are not linked to biological observables at micro-scale sub RB and EB level. For example, the authors just state that “Rather than an exponential distribution as is common for chemical reactions, we use a gamma distribution, which represents the time before a given number of events in a Poisson processes, and which better describes the timing between cell transitions” without providing any justification.

We have now included two literature references to better justify this choice of distribution for the time between cell divisions. While some distributions, such as exponential, can likely be ruled out given experimental data, it would be quite difficult to differentiate a gamma distribution, say, from a sometimes very similar lognormal distribution. In the future, the time between cell divisions could be more narrowly defined by using longitudinal time experiments of individual chlamydiae, which we don't have available for this work.

Also, it is not clear how parameters can be determined using experimental observations for the normal distribution used for determining the growth rate in the context of the cell size dynamics.

We have included a detailed derivation for the mean cytoplasmic growth rate μ_R . The model is actually fairly sensitive to this parameter, and the calculated value leads to growth data that is consistent with the experiments. As in the case of the gamma distribution, our data is not granular enough to determine the actual distribution used for the growth rate, so we used independent normal rates for simplicity. The standard deviation σ_R of the distribution was broadly fit to data, as it was made explicit in the new version of the manuscript with Table 1.

Notice that the mathematical analysis of this simple model is actually independent of either the gamma or the normal distribution – it effectively relies only on the fact that these distributions have finite variance and uses the Central Limit Theorem to conclude that the size distribution after m cell divisions is roughly lognormal. This also highlights the difficulty of narrowing down the distribution, since in this case different distributions can lead to very similar dynamics.

It is not clear how the model can be used to test suggestion by the authors that “conversion could be actively promoted in a smaller RB by the higher effective concentration of a positive regulator” since it does not describe molecular mechanisms of conversion. Also, authors suggest that “contact between the RB and the inclusion membrane, which has been proposed to prevent RB-to-EB conversion, could be reduced by the geometry of a smaller RB”. At the same time, their model does not describe in detail shapes of RBs and membranes as well as interactions of RBs with membranes.

The paragraph on possible mechanisms was intended to speculate about how smaller cell size could result in RB-to-EB conversion. The current work was not meant to test these hypothetical positive or negative regulators of conversion that were speculated upon.

To summarize, the paper describes novel potentially important biologically relevant observations. At the same time, proposed oversimplified models do not provide clear explanation of how exactly RBs and host cells could implement feedback-independent mechanism based on the RB size to control the dynamics of RB-to-EB conversion.

We hope that the new and more detailed model will be more appropriate. In each case, we have used the most parsimonious hypotheses to sufficiently describe the data given the available information, and we believe that the resulting model has an appropriate balance of simplicity and consistency with experimental results. Overall, our goal is to make the case for the idea that RB size control could be sufficient for feedback-independent control of RB-to-EB conversion, a conceptual advance that will hopefully spur future work by our group and others to examine the specific mechanisms.

REVIEWERS' COMMENTS:

Reviewer #2 (Remarks to the Author):

The modifications to the manuscript were appropriate and effective - specifically, the inclusion of binary division v polar budding analysis. There are still a couple of spelling errors (e.g. new figure about division 'dparental'). The analysis and use of parent/daughter is a tad unclear. Dividing cells were identified based upon clear septa? with two nascent daughter cells - therefore, it seems that ratio between daughter cells (vs polar/budding division with a clear parent) would be more appropriate?

Reviewer #3 (Remarks to the Author):

The authors satisfactory responded to most of my comments.
One of the their responses indicates that:

"The paragraph on possible mechanisms was intended to speculate about how smaller cell size could result in RB-to-EB conversion. The current work was not meant to test these hypothetical positive or negative regulators of conversion that were speculated upon".

If they were to clearly emphasize in the Introduction and relevant sections of the paper that they just speculate about mechanisms of how smaller cell size could result in RB-to-EB conversion, I could recommend this paper for publication.

NCOMMS-17-14188B

“Replication-dependent size reduction precedes differentiation in *Chlamydia*” by Lee et al.

Responses to the comments from the editor and the reviewers.

1. We have reformatted the manuscript to conform to the guidelines. In particular we have added a 150-word Abstract, revised the Introduction and added subheaders to the Results.
2. In response to Reviewer 2, we have rewritten the section on RB binary fission to clarify our approach. The reviewer suggested comparing the size ratio of the 2 daughters in a dividing RB. However we have explained that we have taken the approach used in *E. coli*, in which the daughter/parent ratio was calculated for each daughter. The advantage of the daughter/parent ratio is that a histogram of the ratios for a population of dividing RBs reveals whether cell division is polarized and the results can be statistically analyzed with the D’Agostino-Pearson test. Both these analyses are presented in the manuscript.
3. Reviewer 3 asked that we make clear that we are just speculating about mechanisms by which smaller cell size could result in RB-to-EB conversion. We have already used the word “speculate” (p12, line 9) in the relevant paragraph. We are open to any other suggested wording to make clear that this paragraph is speculative.